# Analyzing the Mechanism of Zinc Oxide Nanowires Bending and Bundling Induced by Electron Beam under Scanning Electron Microscope Using Numerical and Simulation Analysis

**DOI:** 10.3390/ma15155358

**Published:** 2022-08-03

**Authors:** Basma ElZein, Ali Elrashidi, Elhadj Dogheche, Ghassan Jabbour

**Affiliations:** 1Department of Electrical Engineering, University of Business and Technology, Jeddah 21432, Saudi Arabia; a.elrashidi@ubt.edu.sa; 2Department of Engineering Physics, Alexandria University, Alexandria 21544, Egypt; 3Institute of Technology (IUT), Université Polytechnique Hauts-de-France (UPHF), 59313 Valenciennes, France; elhadj.dogheche@uphf.fr; 4School of Electrical Engineering and Computer Science, University of Ottawa, 800 King Edward Ave., Ottawa, ON K1N 6N5, Canada; gjabbour@uottawa.ca

**Keywords:** ZnO nanowires, FDTD, SEM method, light absorption, NWs deflection, Au nanoparticles

## Abstract

The bending effect of self-catalyst zinc oxide nanowires on a photoconducting behavior has been investigated by in-situ scanning electron microscope method and interpreted by analytical modeling. Zinc oxide NWs tend to incline due to geometric instability and because of the piezoelectric properties, which was confirmed by scanning electron microscope images. A cantilever bending model adequately describes the bending and bundling events, which are linked to the electrostatic interaction between nanowires. The light absorption of zinc oxide nanowires in the visible and near infrared bands has been modelled using the finite difference time domain method. The influence of the density of nanowires (25%, 50%, 75%) and the integration of plasmonic nanoparticles distributed on the seed layer (with varied radii) on the light absorption of zinc oxide nanowires was studied using simulation analysis. We have shown that the geometry of zinc oxide nanowires in terms of length, separation distance, and surface charge density affects the process of zinc oxide nanowires bending and bundling and that absorption will be maximized by integrating Au plasmonic nanoparticles with a radius of 10 nm.

## 1. Introduction

Zinc oxide nanowires, a one-dimensional (1D) ceramic semiconductor nanostructure, have been widely studies due to their wide application in optoelectronics, solar cells, piezoelectric, and sensing devices [1,2,3,4]. As an abundant, non-toxic metal oxide semiconductor with a relatively wide bandgap of 3.37 eV and a high exciton energy of 60 mev, such material is promising for many optical devices [5,6]. Furthermore, the wurtzite structure and the high piezoelectric coefficient of (ZnO) confers its piezoelectric and polar character along the c-axis direction. Zinc oxide nanowires (ZnO NWs) have been considered as building blocks in a wide variety of devices in the fields of electronics, optoelectronics, photovoltaic, and sensing [7,8].

In addition, (ZnO NWs) can be easily grown on a (ZnO) seed layer (SL) by using a vast number of catalyst-free physical and chemical deposition techniques, including thermal evaporation, pulsed laser deposition, chemical vapor deposition, and spray pyrolysis [9].

A special emphasis is placed on the effect of polarity on the nucleation and growth mechanisms, defect incorporation and doping contact properties, and device performance. Polarity is related to the direction of the cation–anion bond, collinear to the c-axis [10].

So, when the growth starts from a zinc atom to an oxygen atom, the direction is (Zn) polar and vice versa. Pulsed laser deposition (PLD) offers a unique opportunity to select the (O) or (Zn) polarity of the resultant nanowires and its further compatibility with the fabrication processes of flexible devices [4]. The control and use of polarity in (ZnO NWs) grown by (PLD) opens a new way to greatly enhance the performance of related piezoelectric devices. (NWs) often adopt specific orientation during their growth despite the lack of epitaxial guidance from the substrate [11,12]. Although this tendency is not unique to (NWs), it has become more visible in (NWs), with (ZnO) providing the most evidence. [13].

Due to their polarity and elasticity, bent and bundled (ZnO NWs) were induced by electron beams in scanning electron microscopy and have been subject to many interpretations. Hence, the performance of the bent (ZnO NWs) were analyzed by simulation where the absorption property is the main factor that affects the device performance. To enhance the absorption of the structure, plasmonic (NPs) are distributed over the (ZnO) layer, which were used in the visible and infrared regions [14,15,16]. Many applications, such as a highly sensitive biosensor, high absorber and high-efficiency solar cells, use (Au NPs) spread over (ZnO) nanowires and the (ZnO) layer [17,18]. The main parameters that affect the absorption are the NPs shapes, materials used, and surrounding medium [19].

Field emission (FE) is one of the most important applications since the emitting efficiency can be highly improved by the alignment [20]. Comprehensive theoretical and experimental research on (FE) has been mainly conducted on carbon nanotubes (CNTs), owing to their good conductivity; chemical stability; and easy, as well as cost-effective, fabrication. Oxide semiconducting (NWs), which are more stable at high temperatures in an oxygen environment and have a more controllable electronic property, have been considered more and more as alternative (FE) sources instead of (CNTs). With a large exciton binding energy and high melting temperature, (ZnO NWs) have recently been studied as an effective (FE) source. However, very little research has been performed on optimizing their (FE) property by controlling the density of the aligned (ZnO NWs). On the other hand, the bending and bundling of the (ZnO NWs) might adversely affect the transport of electrons in a photovoltaic device.

In this work, a self-catalyst (ZnO NWs) grown on a thin glass/indium tin oxide (ITO) substrate coated with a thin seed layer of (ZnO) is carried out experimentally. The bending and bundling mechanism of the ZnO NWs is examined by scanning electron microscope (SEM) images and is explained analytically using Euler–Bernoulli model. The light absorption of the bent (NWs) is simulated using Lumerical electromagnetic solver and introduced as a function of the incident wavelength in the visible and near infrared (NIR) regions. To enhance the absorption intensity, plasmonic (NPs) of various materials and radii are integrated and distributed uniformly over the (ZnO) layer. In addition, a parametric study has been conducted in terms of ((NW) length, radius, distance between (NWs) and surface charge density of the (Zn^+^) or (O^−^) charges) to analyze the total deflection of the (ZnO NWs).

## 2. Zinc Oxide Nanowires Fabrication (Experimental)

Glass slides coated with thin film of both (ITO) (75 nm) and (ZnO SL) of 400 nm thickness [21] were used as substrates for subsequent growth of (ZnO NW) array with (*L*) as length and d as the average distance between two adjacent (NWs).

(ZnO) seed layer was deposited by (PLD) [22,23]. (ZnO NWs) were grown at 10 Torr Argon environment and temperature less than 500 °C, using laser pulse energy of 400 mJ and target–substrate distance of 6.5 cm [21]. The grown (ZnO NWs) were then characterized by (SEM).

The electrostatic force between (NWs) due to (Zn^+^) and (O^−^) charges on the top of the NWs are illustrated on both NWs (*a)* and (*b)* as (*F_a_*) and (*F_b_*), respectively, as illustrated in Figure 1a, while the bending of the free terminals of the (NWs) is shown in Figure 1b, where the separation distance between the top of the (NWs) has vanished.

## 3. Zinc Oxide Nanowires under Scanning Electron Microscope

We speculate in Figure 2, that some of the (ZnO NWs) tend to incline due to geometric instability during scanning electron microscope (SEM) characterization. In this study, a Carl Ziss AG-Ultra 55 SEM with 7 kV and an energy selective backscattered detector (EsB) were used (Oberkochen, Germany). Since (ZnO NWs) have piezoelectric properties, this causes the side surface to be either positively or negatively charged. Bundling was noticed from vertically aligned (ZnO NWs). Electrostatic interactions owing to charged (0001) polar surfaces or electron beam bombardment during (SEM) could be the cause. (ZnO NWs) grown along the c-axis of a wurtzite structure can be described schematically as the alternative stacking of planes of fourfold coordinated (O^2−^) and (Zn^2+^) ions, and thus flat top surfaces of (ZnO) 1D nanostructures are either (Zn^+^) or (O^−^) terminated (0001) surfaces. Figure 2a shows the (SEM) image of (ZnO NWs) before bending (inset—some highlighted NWs). Figure 2b illustrates the (SEM) image after bending of the (NWs) (inset—highlighted bent and bundled (ZnO NWs)).

The (XRD) spectrum of the (ZnO NWs) grown on glass (ITO) substrate is presented in Figure 3. It is clearly seen the high peak of (ZnO) (002) revealing the dominant c-axis orientation. Different peak positions of the band edge emission in the ultraviolet (UV) region (379.5 nm), as well as defect-induced emission in the visible region: green (531 nm) and yellow (584.2 nm) that were clearly depicted in the photoluminescence (PL) spectrum are shown in Figure 4.

The uncompensated (Zn^+^) and (O^−^) terminated surfaces result in net positive and negative charges, respectively. The bending and bundling of (NWs) were more clearly observed from the (SEM) images. Many (NWs) were bent and attracted to each other at, or near, their tips to form bundles of various numbers of (NWs). Considering the (NW) separation over a few hundred nanometers, it was unlikely that the Van der Waals interaction was responsible for the observed attraction. Moreover, it was found that some of the (NWs) in proximity did not contact each other but bent in such a way to make contact with ones farther apart to form a crossed configuration, as shown in Figure 2b. This indicates that repulsive interaction was occurring in addition to attractive interaction. Presumably, the opposite charges of (Zn^+^) and (O^−^) terminated hexagonal facets of (NWs) were the origin of these interactions [24].

## 4. Modeling and Simulation Analysis of the Fabricated Structure

The effect of bent (ZnO NWs) on light absorption in the (UV) and (NIR) areas, as well as boosting (UV) absorption by dispersing plasmonic (NPs) on the (ZnO) layer, were investigated. In addition, we studied the influence of (NWs) length surface charge density on total deflection of the (NWs).

### 4.1. Numerical Analysis

In this section, the analytical model that explains the main reason behind the bending and bundling mechanism of (ZnO NWs) under electron beam bombardment is introduced. The bending of the (NWs) increases as the positive and negative charges on the (NWs) free terminals increases, which leads to an increase the electrostatic force on the nanowire free ends (*F*). Hence, bending (*B*) occurring in a given (NW) can be calculated using the Euler–Bernoulli Equation [24] as illustrated in Equation (1).
(1)$$B=\frac{FL^{3}}{3EI}$$
where (*E*) is the modulus of elasticity for (ZnO) material, (*L*) is the nanowire length, and (*I*) is the moment of inertia, which is given by Equation (2) [24], and can be calculated using Coulomb’s law, as shown in Equation (3) [24]:(2)$$I={\left(\frac{5\sqrt{3}a}{16}\right)}^{4}$$
where (*a*) is the radius of the (NWs). To calculate the moment of inertia, the (NW) is considering as a cantilever with a fixed end at the (ZnO) base and free end at the (NW) top end.
(3)$$F=\frac{1}{4\pi \varepsilon_{0}\varepsilon_{r}}\frac{Q_{a}Q_{b}}{d^{2}}$$
where (*Q_a_*) and (*Q_b_*) are the point charges on both (NWs) (*a*) and (*b*), (*d*) is the distance between two (NWs), (*ε*_0_) is the dielectric constant of the air, (*ε_r_*) is the relative permittivity.

Hence, the contact between nanowire (*a*) and nanowire (*b*) can occur when the bending in both nanowires is greater than the separation between them and satisfies the following condition (Equation (4)):(4)$$B_{a}+B_{b}\geq d$$
where (*B_a_*) is the bending in nanowire (*a*) and (*B_b_*) is the bending in nanowire (*b*) and the total bending of both wires should be equal or greater than the separation distance (*d*) between two (NWs), so we are sure that they will touch each other.

Given the high aspect ratios of the nanowires, the tips can be considered as a point charge (*Q*) as illustrated in Equation (5). Hence, the forces action on both nanowires are given in the following Equation (6) [20]:(5)$$Q=\left(\frac{3\sqrt{3}}{2}\right)\times a^{2}\times \sigma$$
(6)$$F_{a}=F_{b}=\frac{1}{4\pi \varepsilon_{0}\varepsilon_{r}}\frac{Q^{2}}{d^{2}}=\frac{1}{4\pi \varepsilon_{0}\varepsilon_{r}}\frac{27\;a^{4}\;\sigma^{2}}{{\left(2\;d\right)}^{2}}$$
where (*σ*) is the surface charge density (C/m^2^).

Plasmonic nanoparticles distributed on the (ZnO) layer is changing the absorbed optical power inside, which depends on the maximum value of reflectivity. (NP) shape is the main parameter on the transmitted optical power, as well as the relative permittivity of the plasmonic (NPs) and dielectric constant of the surrounded medium [25]. The maximum absorption occurred at the maximum value of the wavelength, which can be calculated using Equation (7).
(7)$$\lambda_{max}=\frac{P}{n}{\left(\frac{\varepsilon_{P}\varepsilon_{m}\left(\lambda_{max}\right)}{\varepsilon_{m}+\varepsilon_{P}\left(\lambda_{max}\right)}\right)}^{1/2}$$
where (*ε_m_*) is the permittivity of the surrounding medium, (*ε_P_*) is a plasmonic (NP) dielectric constant at corresponding (*λ_max_*), (*n*) is an integer, and (*P*) is structural periodicity.

Hence, the dielectric permittivity can be expressed by using a multi-oscillator Drude–Lorentz model [25], as given in Equation (8):(8)$$\varepsilon_{plasmonic}=\varepsilon_{\infty }-\frac{\omega_{D}^{2}}{\omega^{2}+j\omega \gamma_{D}}-\overset{6}{\underset{k=1}{\sum }}\frac{\delta_{k}\omega_{k}^{2}}{\omega^{2}-\omega_{k}^{2}+2j\omega \gamma_{k}}$$
where (*ε_∞_*) is the plasmonic high frequency dielectric permittivity, (*ω_D_*) and (*γ_D_*) are the plasma and collision frequencies of the free electrons, (*δ_k_*) is the amplitude of Lorentz oscillator, (*ω_k_*) is the resonance angular frequencies and (*γ_k_*) is the damping constants for (*k*) value from one to six.

### 4.2. The Effect of Nanowire Length and Surface Charge Density on the Total Nanowire Deflection

The main parameters that affect the total (NWs) deflection, which leads to their bending, are calculated using Equations (1)–(6) as a function of (NWs) length and surface charge density distributed. Figure 5a shows the total deflection of the (NWs) as a function of (NWs) length ranging from 300 nm to 550 nm at a fixed value of surface charge density of 0.022 C/m^2^, using three different values of (NWs) separation, *d* = 40 nm, *d* = 50 nm, and *d* = 60 nm. In this regard, the total deflection increases as the (NW) length increases. Longer (NWs) are free to bend more, with the total deflection shifting upwards for smaller separation distance of 40 nm between the (NWs), as illustrated in the red curve in Figure 5a. As shown in Figure 5a, any value of (NW) length will satisfy the condition given in Equation (4) and the (NWs) will contact each other, where the total deflection will be higher than the (NWs) separation.

At 50 nm separation, the total deflection for surface charge densities 0.01, 0.022, and 0.06 C/m^2^, respectively, are shown in Figure 5b. By applying Equation (4), we can conclude that, the total deflection for *σ* = 0.001 C/m^2^ cannot satisfy the deflection condition, thus preventing the bending of (NWs) toward each other, as illustrated in the insert of Figure 5b. The minimum value of the total deflection in this case is 50 nm. Using the same insert, for σ = 0.01 C/m^2^, the length should be more than 365 nm to satisfy the deflection condition as illustrated in the sub Figure 5b. The subfigure in Figure 5b shows the condition that will not satisfy the contact condition of (NWs) in Equation (4), where the total separation distance is 50 nm.

In addition, the total deflection as a function of the surface charge density is illustrated in Figure 6a for different (NWs) lengths of 300 nm, 400 nm, and 500 nm, at 50 nm, at a fixed (NWs) separation of 50 nm. The total deflection increases as the surface charge density increases. There is no deflection for (σ) less than 0.005 C/m^2^, as shown in Figure 6a. Furthermore, the total deflection as a function of surface charge density for different separation distance of 50 nm, 60 nm, and 70 nm, respectively, and a fixed (NWs) length of 400 nm, is shown in Figure 6b. The minimum surface charge density to satisfy the condition in Equation (4) is 0.006 C/m^2^, 0.007 C/m^2^, and 0.014 C/m^2^ for (NW) length 500 nm, 400 nm, and 300 nm, respectively, at separation of 50 nm, as illustrated in Sub-Figure 6a. Furthermore, the minimum surface charge density to satisfy the condition in Equation (4) is 0.008 C/m^2^, 0.012 C/m^2^, and 0.015 C/m^2^ (NW) for (NWs) separation 50 nm, 60 nm, and 70 nm, respectively, at (NW) length 400 nm, as illustrated in Sub-Figure 6b.

### 4.3. Absorption in the Visible and Near Infrared Regions

A simulation analysis of the fabricated structure is introduced in this section to calculate the absorbed light in the visible and (NIR) regions. The total absorption of uniformly distributed (ZnO) nanowires over a (ZnO) seed layer is calculated using the Lumerical software package, finite difference time domain method (FDTD), with a mesh size of 0.25 nm, periodic boundary conditions in both x and y dimensions, and a perfect matched layer in the z direction. The incident spectrum has a wide range, 0.4 µm to 2.6 µm, to cover the visible and (NIR) regions with an incident plane wave source in the z direction. Figure 7 shows the absorbed light (%) for straight and bent (NWs). To be consistent with the experimental values, the length of the (NW) is assumed to be 400 nm with a radius of 60 nm, the spacing between two successive (NWs) is taken to be 50 nm, and the distance between the centers of (NWs) is 110 nm. We consider a unit cell of 240 × 240 nm^2^ with two bent (ZnO NWs) with a periodic structure in both the x and y directions in the simulation design.

As illustrated in Figure 7a, the absorption in visible region differs from the absorption in the NIR region, while more peaks are detected in the visible region, as in Figure 7b. The absorption in the visible region is very limited and has only one peak at 420 nm for straight (ZnO NWs). However, the absorption of the bent (NWs) is increased in the same wavelength region and has many peaks in the visible band. Alternatively, the absorbed light in the (NIR) region, 0.7–2.6 µm, is high and almost identical for straight and bent (NWs). As a result, once the peaks arise in the visible area, we may detect the presence of bent (NWs).

### 4.4. Effect of Bent Nanowires Concentration on the Absorpance

The concentration of the bent (NWs) in the same unit cell is playing an important role in the absorption process, hence the effect of changing the number of bent (NWs) in the same unit cell is studied. The surface area of the unit cell is taken to be 680 × 680 nm^2^ and the diameter of the (NW) is 120 nm with a separation between them of 50 nm. In this sense, we assume a uniform distribution of the (NWs) with 12 (NWs) in the unit cell. Figure 8 shows the absorption of the (ZnO NWs) in the unit cell with different concentration of the bent (NWs). The simulated concentrations for bent (NWs) are 25%, 50%, 75%, respectively. The maximum absorption is obtained when the bending concentration is 50% (the remaining 50% of the (NWs) in the unit cell is straight), as shown in Figure 8. The absorption is increased with increasing the number of bent (NWs), which can be used as a good absorber.

### 4.5. Enhancing the Absorption Using Plasmonic NPs

The surface plasmon localization (SPL) technique on metallic (NPs) is used as one of the most efficient methods to improve the optical absorption in solar cells and optical sensors [26,27]. In this regard, we distribute plasmonic (Au NPs) on the top surface of (ZnO) base layer (seed layer), shown in Figure 9, to enhance the absorption in the (UV) region [28].

The absorption for bent (ZnO NWs) without (NPs) and with (Au NPs) having a radius (R) of 10 nm, 20 nm and 30 nm, respectively, is shown in Figure 10. The absorption for R = 10 nm is enhanced in the visible region and in the (NIR) region, from 2.2 µm to 2.6 µm. The maximum absorption obtained at 1640 nm before adding (Au NPs) is shifted to 1570 nm for the 10 nm (Au NPs), while the intensity decreases from 78% (no NPs) to 61% (with NPs). On the other hand, the maximum absorption is 1650 nm for 20 nm (Au NPs) and shifted to 1580 nm for the 30 nm (Au NPs), while the intensity decreases from 88% (20 nm NPs) to 67% (30 nm NPs). (Au NPs) behave like a nanoantenna, which absorb the incident light and retransmit it into the structure, where peak wavelength of the absorbed light is depending on the (Au NPs) size and periodicity as given in Equation (7). Changing the plasmonic material will change the peak maximum wavelength to different range according to the Drude–Lorentz model [25]. For (Au NPs), the maximum peak occurs at 400 nm in addition to other shifted peaks, according to the (Au NPs) radius as illustrated in Figure 10.

## 5. Conclusions

The bending of (ZnO NWs) phenomena observed under electron beam bombardment, as confirmed by (SEM) images, is due to (NWs) geometric instability. Then, the main factors that affect the (NWs) total deflection are analyzed numerically for different (NWs) length, (NWs) separation, and surface charge density. The surface charge density, (NWs) separation and (NWs) length are affecting the required bending and bundling of the (NWs). Hence, the total (NW) deflection at (NWs) length 300 nm, 400 nm, and 500 nm at 50 nm separation between two (NWs) as a function of the surface charge density was illustrated in this work. The minimum values of surface charge densities that allow contact between (NWs) and satisfy the bending condition are 0.006 C/m^2^, 0.007 C/m^2^, and 0.014 C/m^2^ for (NWs) length 300 nm, 400 nm, and 500 nm, respectively, at (NWs) separation 50 nm. Moreover, the minimum surface charge density to satisfy the contact of (NWs) is 0.008 C/m^2^, 0.012 C/m^2^, and 0.015 C/m^2^ for (NWs) separations 50 nm, 60 nm, and 70 nm, respectively, at (NW) length 400 nm. Furthermore, the light absorbed by the bent (NWs) is simulated in the visible and (NIR) regions by using (FDTD) method and Lumerical software package. The absorption is enhanced in the (UV) region and almost not changed in the (NIR) region. Finally, the absorption was enhanced by using plasmonic (NPs) with different radii distributed on the seed layer. In this case, maximum absorption is obtained for the case of (Au NPs) with 10 nm radius, where the enhancement occurs in both visible (40%) and (NIR) regions (20% between 2.2 µm to 2.6 µm). The maximum absorption obtained at 1640 nm and 1570 nm without and with (Au NPs) of radius 10 nm.

## Figures and Tables

**Figure 1 materials-15-05358-f001:**
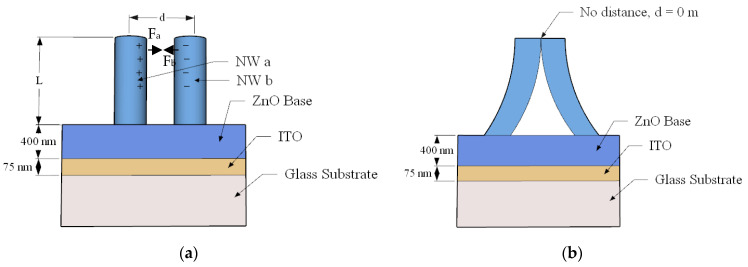
Fabricated structure (**a**) straight ZnO NWs and (**b**) bent ZnO NWs.

**Figure 2 materials-15-05358-f002:**
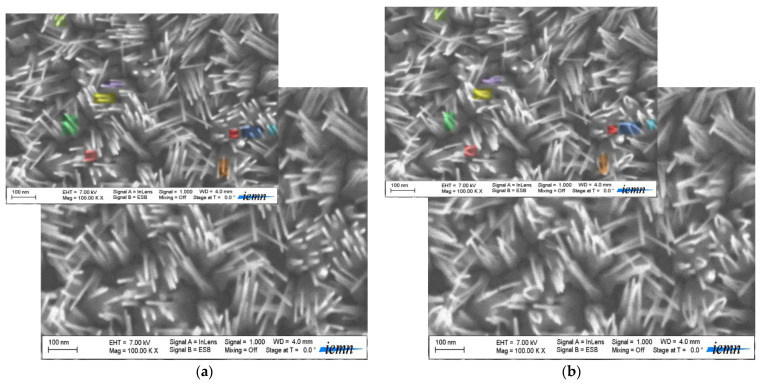
SEM Images top view of ZnO NWs grown at 10 Torr in Argon environment, 400 mJ for 15 min. (**a**) ZnO NWs by PLD on thin Glass + ITO + ZnO SL (before bending); (**b**) ZnO NWs by PLD on thin Glass + ITO + ZnO SL (after bending).

**Figure 3 materials-15-05358-f003:**
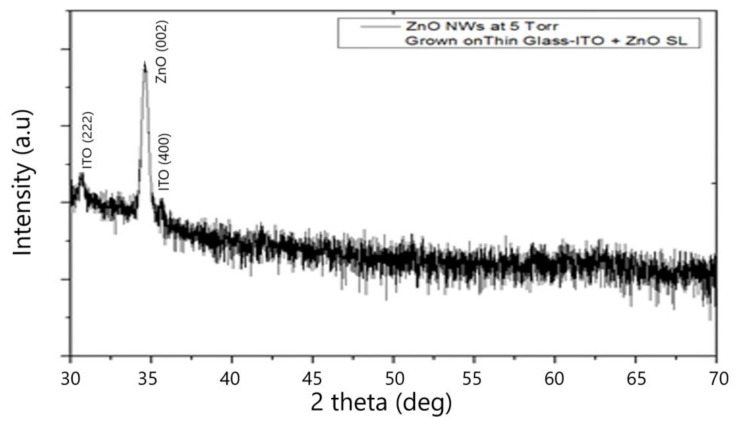
XRD Spectrum of ZnO Nws grown on Glass-ITO + ZnO SL at 10 Torr, Tsub < 500 °C d target–substrate = 6.5 cm, JCPDS card for ZnO file number: 070-8070.

**Figure 4 materials-15-05358-f004:**
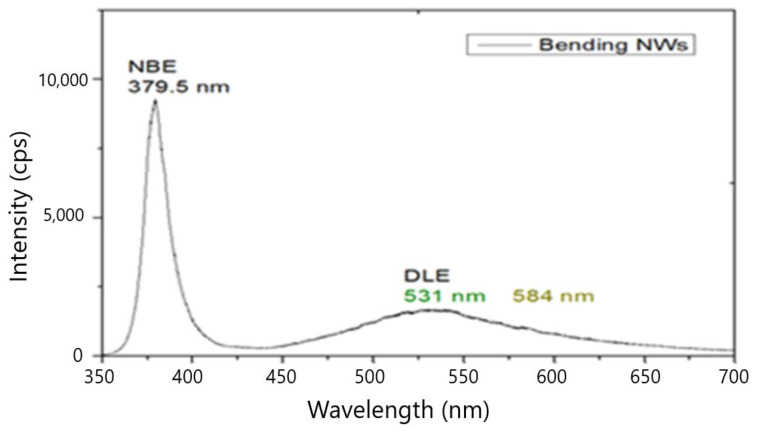
PL Spectrum of ZnO Nws grown on Glass-ITO + ZnO SL at 10 Torr, Tsub < 500 °C d target–substrate = 6.5 cm.

**Figure 5 materials-15-05358-f005:**
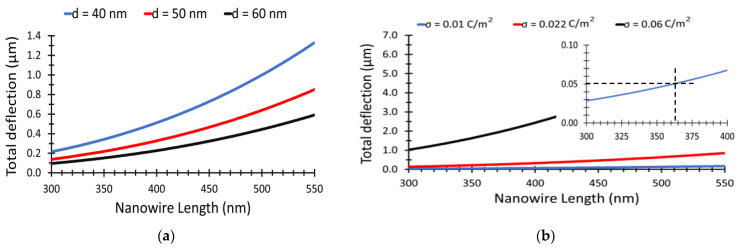
Total deflection versus NWs length (**a**) for different NWs separations, (**b**) for different surface charge density.

**Figure 6 materials-15-05358-f006:**
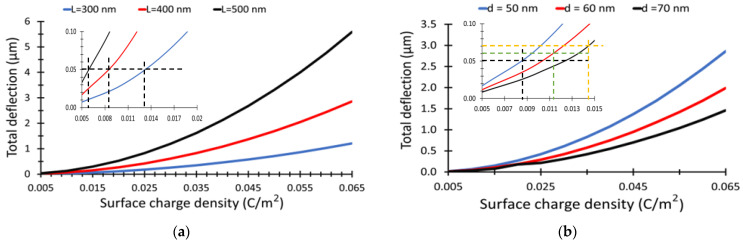
Total deflection of the NWs versus NWs surface charge density, (**a**) for different NWs length at a fixed separation between NWs of 50 nm, (**b**) for NWs separation of 50 nm, 60 nm, and 70 nm, respectively, and a fixed length of 400 nm.

**Figure 7 materials-15-05358-f007:**
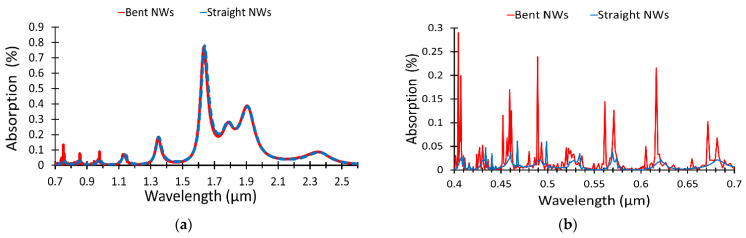
Light absorption vs. wavelength for straight and bent NWs (**a**) at NIR (**b**) visible bands.

**Figure 8 materials-15-05358-f008:**
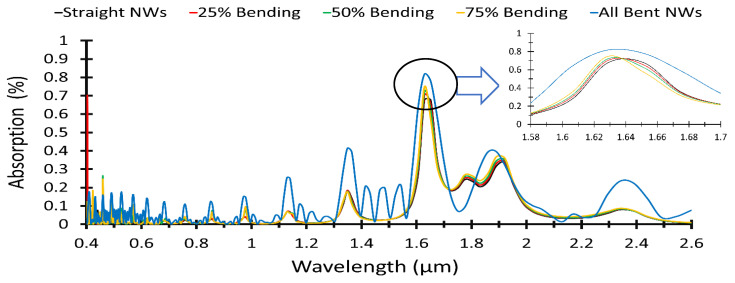
Light absorption for different concentration of bent NWs ranging from 0% (no bending) to 100% (NWs in the given simulation cell are all bent).

**Figure 9 materials-15-05358-f009:**
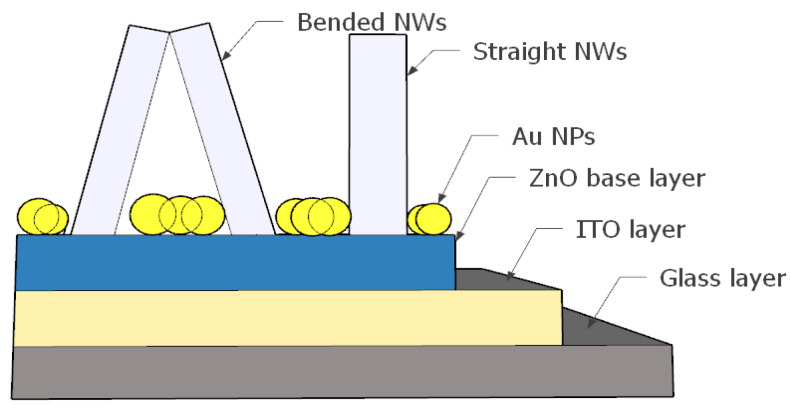
Proposed structure with Au NPs distributed on the top of the ZnO layer.

**Figure 10 materials-15-05358-f010:**
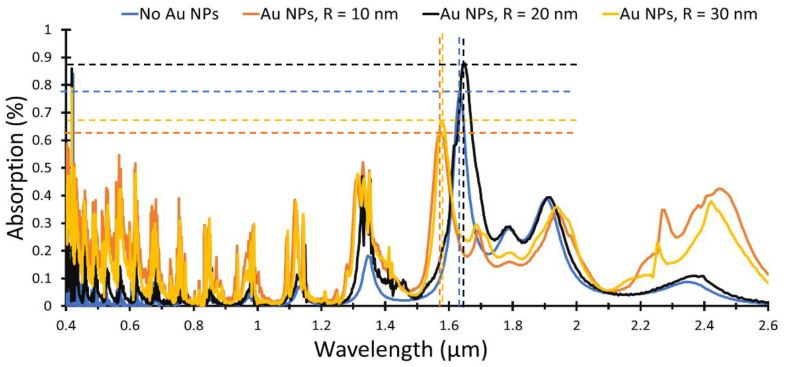
Absorbance of the proposed bent ZnO NWs without and with Au NPs of radius 10, 20 and 30 nm, respectively, distributed on the top of the ZnO base layer.

## Data Availability

Not applicable.

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
