# Peer review of "Analyzing the Mechanism of Zinc Oxide Nanowires Bending and Bundling Induced by Electron Beam under Scanning Electron Microscope Using Numerical and Simulation Analysis"

_materials, 2022, doi:10.3390/ma15155358_

Round 1
Reviewer 1 Report
1. Please pay more attention to the format requirement of the journal, uniform the fonts at least.
2. The resolution of all the figures need to be improved significantly, again, please check the requirement of the journal again for the figures.
3. This article is lack of innovation, and cannot find any practical guidance.
4. Line 12, when "ZnO NWs" firstly appear, the full name of "NWs" is more preferred.
5. Line 51, "So, ...and vice versa." Is it a new paragraph or not?
6. Line 216, the equation is needed to be centered.
7. The conclusion need to be re-organized, make it more profound.
Author Response
Kindly check the attached file

Reviewer 2 Report
There are some questions and problems arose that requires correction and /or explanations:
1. The sentence “the integration of plasmonic material (with varied radii) on optimizing the light absorption of ZnO NWs was studied using parametric research using numerical analysis.” Is not well written in English.
2. The inset picture in Figure 2 are too to see. It is better to enlarge the inset pictures instead of the large picture below.
3. The ordinate in figure 3 does not need scale label, because it used arbitrary unit. The legend ZnO (002) obscures the spectral line. The tagged position of ZnO (100) is not suitable. Figure 3 need redraw.
4. What the B and L stand for in Equation (1)?
5. Why the forces action on both nanowires are different from Equation (5a and 5b) , it is seem that two nanowires are indistinguishable.
6. Line 194, there are a typo, Figure 5 should be Figure 7.
7. Page 7, the figures 7 does not show the UV band, but how conclude “more peaks are detected in the UV region”?
8. From figure 10, The maximum absorption obtained at 1650 nm and 1580 nm should be with Au NPs of radius 30 nm and 20 nm, why the authors think “The maximum absorption obtained at 1650 nm and 1580 nm without and with Au NPs of radius 10 nm without and with Au NPs of radius 10 nm”?
Author Response
Kindly check the attached file

Reviewer 3 Report
Could this research work have any effect on the technological development of institutions or countries?
What is your contribution, with this research work, to the academic-scientific community of sciences and engineering?
How and why can your research work contribute to the strengthening of science, engineering, and technological development of a particular society or scientific community?
Which is the environmental, social, and economic impact of your research work or study make?
The grammar could be improved; some grammatical errors are present. I consider that this manuscript must be reviewed by a person who specialized in grammar English.
I consider that the author must report a concrete, concise and coherent title according to the results that are informed; likewise, not report acronyms or symbols in the title, the acronyms should define in other sections (i.e., introduction, methods, results, and discussion). I suggest modifying the paper title.
Which numerical and simulation analyses are made?
Where are the details of the numerical and simulation analysis from SEM?
Where are the details of the model FDTD of the light absorption in the NIR?
Why do you use NWs density of 25%, 50%, and 75%?
Why is the absorption maximized with Au NPs of 10 nm? ¿What occurs with other sizes of radii? What occurs with another type of element different from Au? I think that using Au NPs is an obvious result, where is the new or novelty?
What is the SEM Method? Please, explain.
I consider that the acronyms should not be defined in the abstract; the acronyms should define in other sections (i.e., introduction, methods, results, and discussion). Likewise, the acronyms must be defined only once and when they are first presented. I suggest modifying the abstract.
Change this keyword, please use keywords different from the title; you can use synonyms. I suggest changing the keywords.
All names of the compounds or chemical formulas must be reported correctly with the symbol or acronyms between brackets. I suggest reviewing all the writing of the manuscript; after you define all the compounds or chemical formulas you can use the symbol or acronyms in your writing.
I suggest not reporting symbols or acronyms in the titles or subtitles; I suggest modifying all the titles or subtitles that have symbols or acronyms.
In line 27, you must report so: Zinc oxide nanowires (ZnO NWs); all the acronyms must be between brackets.
In line 36, you must report so: zinc oxide (ZnO); all the acronyms must be between brackets.
In line 42, check paragraph indentation.
In line 43, you must report so: pulsed laser deposition (PLD); all the acronyms must be between brackets.
In line 43, you must report so: oxygen (O); all the symbols must be between brackets.
In line 44, you must report so: zinc (Zn); all the symbols must be between brackets.
In line 47, you must report so: nanowires (NWs), all the acronyms must be between brackets.
In line 54, you must report so: nanoparticles (NPs), all the acronyms must be between brackets.
In line 71, you must report so: glass/indium tin oxide (ITO)
In line 76: Which is the name? Defines this acronym.
You must include Figure 1 after the Paragraph of lines 90-93
You must report so the Figure 1 caption: (a), (b); the letters between brackets.
In line 94, you must modify the subtitle.
In line 96, you must report so: scanning electron microscope (SEM)
In line 97, you must report so: (EsB), has a bracket of more.
In line 104, you must report so: Figure 2(a); and line 105 must be: Figure 2(b). I suggest checking the manuscript and reporting correctly in the paragraphs when is mentioned all Figures (i.e., Figure 1(a), Figure 1(b), Figure 2(a), Figure 2(b), ..., etc.)
The Figure 2 caption, you must report so: Figure 2. SEM Images top view of ZnO NWs gorown at 10 Torr in Argon environment, 400 mJ for 15 minutes. (a) ZnO NWs by PLD on thin Glass + ITO + ZnO SL (before bending); (b) ZnO NWs by PLD on thin Glass + ITO + ZnO SL (after bending).
In line 113: Which is the name UV? Defines this acronym.
You must include Figure 4 after the Paragraph of lines 120-130
In line 127, you must report so: Figure 2(b).
In line 136, you must modify the subtitle.
In line 144, why E is the modulus of elasticity only for ZnO material? You must explain; What occurs with the ITO? Explain.
In line 144, why "I" is calculated with this equation? You must explain the physics and reference system or system geometry for this equation.
In line 145, where is the reference for equation (2)?
In Equation (3), you must define all the terms of the equation; what signifies ε0?
In line 149, you are defining terms of two-equation (Equation (2) and Equation (3)); therefore, I suggest improving the writing in that specify which are terms of Equation (2) and which are terms of Equation (3)
In line 153, you must report so: ... following condition (Equation (4)).
In Equation (4), you must explain the physics and all terms of Equation; moreover, Why do you use this condition? Where is the bibliography reference for this Equation?
In line 155, you must report so: Equation (5a) and Equation (5b). although, I suggest that the Equations number be changed by: Equation (5) and Equation (6), respectively.
In line 160, you must modify the subtitle.
In line 162, you must report so: Equation (5a) and Equation (5b).
In line 163, possibly the Figure name is incorrect; you are referring to Figure 4(a) or Figure 5(a)? You must check.
According to the last suggestions, you must review, details read your manuscript, and correct all Figure descriptions (i.e., all paragraphs that describe Figure 3, Figure 4, Figure 5, Figure 6, Figure 7, Figure 8, Figure 9, and Figure 10).
In line 165, you must report so: d = 40 nm, 50 nm, and 60 nm.
In line 168, you must report so: Figure 5(a).
I suggest that the behavior reported in Figure 5 must be deepened; you must make a major analysis and discussion of these results.
The Figure 5 caption, you must report so: Figure 5. Total deflection versus NWs length (a) for different NWs separations, (b) for different surface charge density.
In line 171, possibly the Figure name is incorrect; you are referring to Figure 4(b) or Figure 5(b)? You must check.
In line 173, you must report so in the same line: Figure 5(b).
I suggest that the behavior reported in Figure 6 must be deepened; you must make a major analysis and discussion of these results.
The Figure 6 caption, you must report so: Figure 6. Total deflection of the NWs versus NWs surface charge density, (a) for different NWs length at a fixed separation between NWs of 50 nm, (b) for NWs separation of 50 nm, 60 nm, and 70 nm, respectively, and a fixed length of 400 nm.
In line 179, you must report so: 300 nm, 400 nm, and 500 nm.
In line 181, you must report so: Figure 6(a).
In line 183, you must report so: 50 nm, 60 nm, and 70 nm.
In line 184, you must report so: Figure 6(b).
In line 187, you must modify the subtitle.
I suggest that the behavior reported in Figure 7 must be deepened; you must make a major analysis and discussion of these results.
You must include Figure 7 after the Paragraph of lines 202-208.
In line 209, you must modify the subtitle.
In line 212, you must report 680×680 nm2 with the number 2 in superscript.
I suggest that the behavior reported in Figure 8 must be deepened; you must make a major analysis and discussion of these results.
In line 213 and line 214, you report "in this sense, we assume a uniform distribution of the NWs with 12 NWs in the unit cell" Why? How justifies this assumption?
In line 222, you must modify the subtitle.
In line 223, you must report only NPs because you should have defined this acronym in the past sections.
I suggest that the behavior reported in Figure 10 must be deepened; you must make a major analysis and discussion of these results.
The Figure 10 caption, you must report so: Figure 10. Absorbance of the proposed bent ZnO NWs without and with Au NPs of radius 10 nm, 20 nm, 231 nm, and 30 nm, respectively, are distributed on the top of the ZnO base layer.
You must include Figure 10 after the Paragraph of lines 233-237.
In line 235, you must report so: 2.2 μm to 2.6 μm.
I consider that the acronyms should not be defined in the conclusions; the acronyms should define in the other sections (i.e., introduction, methods, results, and discussion). Likewise, the acronyms must be defined only once and when they are first presented.
In line 244, you must report so: 300 nm, 400 nm, and 500 nm.
In line 251, you must report so: 2.2 μm to 2.6 μm.
I recommend checking the references; I consider that there are reference reports with an inadequate template.
Author Response
Kindly check the attached file

Reviewer 4 Report
The authors presented mechanism of ZnO NWs bending and bundling induced by electron beam under SEM using numerical and simulation techniques. They performed a series of experimental and simulations work. The attempted to see the effect of Au nanoparticle on the light absorption properties of bent ZnO NWs. The manuscript is well written and presented with some minor typos, like, Nw etc.
But the effect of bent ZnO is not very clear (as presented in Fig 7) as well as the interpretation is not convincing. A better explanation might help the readers. I recommend the to resubmit the manuscript after improving the explanations.
Author Response
Kindly check the attached file

Round 2
Reviewer 1 Report
Accept
Author Response
Thanks
Reviewer 2 Report
1. what is the meaning of “Uniltra Vilote” ?
2. if UV stand for “ultraviolet”, while , the wavelength region of UV band is 10~380 nm, the figure 7 b shows the 0.4um to 0.7um is visible light band, is not UV band!
3. Why the forces action on both nanowires are different from Equation 5? the constant is 135 and 27 for Fa and Fb, respectively. it is seem that two nanowires are indistinguishable.
4. Figure 4, the unite of ordinate should be “cps”, not “a.u.”
5. when acronyms first appear in the text, the full word should be given. Such as PL in figure 4.
Author Response
- what is the meaning of “Uniltra Vilote” ?
Yes, it is a mistake --- it should be Visible light region.
It is mentioned in the abstract and introduction.
It is corrected, thanks.
- if UV stand for “ultraviolet”, while , the wavelength region of UV band is 10~380 nm, the figure 7 b shows the 0.4um to 0.7um is visible light band, is not UV band!
Yes, it is a mistake --- it should be Visible light region.
It is corrected, thanks.
- Why the forces action on both nanowires are different from Equation 5? the constant is 135 and 27 for Fa and Fb, respectively. it is seem that two nanowires are indistinguishable.
The equations are clear now, Equation (5) is added to clarify the equations.
- Figure 4, the unite of ordinate should be “cps”, not “a.u.”
Yes, it is a mistake.
It is corrected, thanks.
- when acronyms first appear in the text, the full word should be given. Such as PL in figure 4.
It is added, thanks.
Reviewer 4 Report
It has been improved and can be published.
Author Response
Thanks
This manuscript is a resubmission of an earlier submission. The following is a list of the peer review reports and author responses from that submission.
Round 1
Reviewer 1 Report
The present manuscript formally reports on in-situ experimental analysis of the optical performance of ZnO nanowires, but it actually mainly presents simulation results, not compared with, or verified by, real optical measurements. Moreover, authors claim to have investigated the effect of bending and bundling of the NWs (that become binding and bundling in the title) without clarifying if it is an effect due the intrinsic piezoelectric properties, as suggested in the abstract, or to the morphological SEM analysis itself. Furthermore, the enhancement effect associated to the possible addition of plasmonic AuNPs in the NWs forest is only simulated and not even experimentally attempted. Therefore, neither the ultimate goal nor the actual results are clearly declared and expressed.
Moreover, the poor quality of the presentation of both the obtained data and the applied methods is embarrassing. Entire sentences are incomprehensible, some terms are missing, others are inserted incorrectly, some statements make no sense (for the ZnO growth “temperature was less than 500°C”), some parts of the introduction are included at the end of the results and discussion section. The paragraphs are only partially numbered, the captions of the figures are incomplete (see figure 3, which is the significant difference between 3a, 3b, 3c and 3d?), some figures are unnecessarily repeated (see figures 5, 6a and 6b).
In my opinion, the paper in its present form is absolutely not suitable for publication, and perhaps not even for evaluation.
Reviewer 2 Report
El Zein and co-authors studied using SEM, XRD, PL the bending of ZnO nanowires grown on glass indium tin oxide substrate. In order to clarify this effect, they also calculated the light absorption (visible and NIR) of straight and bended ZnO NWs as well as the deflection of the NWs as a function of surface charge density for different separation values.
In Simulation and Analytical Analysis sections the obtained results are described in details, but in my opinion is not focused enough on the conclusions that follow from them.
I think it’ll be better abbreviations like ITO and PLD to be mentioned once in full form: Indium tin oxide and Pulsed laser deposition, respectively.
Some minor issues:
In abstract: „infra-red“ should be “infrared“
On page 4 in the caption of Figure 3, which should be Figure 2, there are some typos – “Imgaes” and “gorwn”.
On page 6, Figure 7 in the legend: “beneded” instead “bended”
On page 9 line 6: furthermore should be with capital letter
On page 9 in conclusions second line: “on the of photo”
Reviewer 3 Report
- What's the purpose or what is the application of the binding and bundling ZnO nanowires? Should be better if mentioned in the title.
- I am confused what is the difference between Figure 3(a)& 3(b), 3(c)&3(d), by the way, the resolution is kind of too low to be publised.
- There are two "Fig. 3" in this mansucript and there is no "Fig. 2". Please re-organized the latter Fig. 3 and other figures, especially when the size of the sub-figures are different, which seems very unprofessional.
- Please re-write the Abstract and Conclusion, should not be so similar, since they have different functions.
- I would like to suggest going through the manuscript more carefully for clarity, syntax and correctness. The English should be improved for the sake of clarity.
Reviewer 4 Report
The authors review the performance analysis of binding ZnO nanowires and study their piezielectric performance via fabrication and simulations. The research idea is well conceived; however, the manuscript has serious flaws and need major revisions. I would like to decide based of the revision of the following comments.
1- The title is very generic and vague. Provide a better descriptive title.
2- Provide the novelty of the study in the abstract or the introduction sections of the manuscript.
3- There are many grammatical and structural errors in the manuscript draft. It needs to be checked by an English editor or a linguistic expert.
4- I recommend the authors to write the manuscript on the MDPI template with details of the experimental procedures and results, simulations and discussions.
5- Where is the experimental section and the ZnO NW fabrication details?
6- Figure 1 is very blur. Provide a better Figure with a relevant description in the manuscript draft.
7- Provide bibliographic reference of the facts narrated on page 3, paragraph one of the results and discussion section.
8- The results are neither descriptive nor argumentative and are not scientifically sound.
9- For clarity, provide a clear and concise scalebar in the SEM micrographs. Also, the relevant discussion in the manuscript draft is very unclear and vague. provide the relevant discussion in the draft with all the appropriate references and each and every aspect of the SEM micrograph should be discussed. What does the yellow, red and blue highlights mean in the Figures?
10- Provide the JCPDS card reference for Figure 3a XRD.
11- How does the crystal defects and career concentration affect the ZnO nanowire polarity and piezoelectric performance?
Round 2
Reviewer 1 Report
Even if authors have taken into account the comments and indications of the first report, partly rewriting the paper, I consider it still not suitable for publication.
The main concern is related to the fact that the work appears incomplete, presenting a mixed experimental and theoretical approach. In fact, authors present the fabrication of the ZnO NWs, and the realized substrates are shown and briefly characterized by SEM, XRD and photoluminescence. Then the bending effect of the NWs after SEM electron beam irradiation is noticed, and the effect of this bending on the optical properties in the UV-VIS-NIR range is simulated. Thus, to correctly complete the study, it would have been necessary to present the experimental absorption spectra, comparing them with the modelling results. Instead, no experimental data are given, which would be important to demonstrate the possible application of the investigated bending and bundling process, but further simulations are added to illustrate the consequences of possibly using metallic nanoparticles distributed on the NWs to obtain enhancement effects. Therefore, the paper does not give real indications for the reader, except for a simple explanation of the NWs bending as due to electrostatic forces.
I suggest to add absorption spectra, before and after SEM imaging, to complete the presentation and obtain real conclusions that could be of actual interest for the reader.
Moreover, the following further points must be considered:
1) An extensive English revision is required
2) The presentation of SEM images in figure 2 is not appropriate. The relevant area and details are difficult to be identified. There is no reason to present a large full image, and just a small inset containing the important selected regions. Please consider the possibility to show only the two enlarged insets. Furthermore, there are substantially three separate captions under the figure, and none of them are clear and contain all the information relevant to immediately understand and interpret the differences between the right and left side. The figure must be “self-standing”, and have a single caption, in which the a and b panel are described and the eye is guided towards the relevant details. Similarly, a single caption must be given for figure 3.
3) Given that, as stated by the authors, the bending and bundling effect is caused by the electron beam irradiation during the realization of the SEM image itself, which is the reason why it is only visible in figure 2b and not in figure 2a?. The two SEM acquisitions are clearly taken exactly on the same area, and I suppose that they are acquired one after the other. Thus, the bending effect under irradiation is not immediate, but it takes some time to be activated? Both the timing of the acquisitions and of the effect must be further discussed, because this is an essential point for applications and to demonstrate the possible control of process also in terms of bended NWs concentration, as simulated afterwards in the paper.
4) The entire paragraph from “Field emission (FE) is one of the most important…” to “might adversely affect the transport of electrons in a photovoltaic device” must be moved from the results section to the introduction, because it defines the context and significance of the work.
Author Response
Kindly check the attached file

Reviewer 3 Report
Accept
Author Response
Thanks
Reviewer 4 Report
The manuscript is still fraught with many problems and need serious attention. The major comments in the revision round 1 are not addressed at all. The manuscript is not in accordance to the professional standards of materials and is not acceptable in the present form.
1- The manuscript is still not written on MDPI's template.
2- The provided line numbers for the response are wrong because of the track changes into the draft file. The authors should take care of such petty mistakes to help the volunteer reviewers, which ultimately benefit the authors.
3- Provide the fabrication process into the manuscript.
4- The results are still neither descriptive nor argumentative and are not scientifically sound. More specifically, I do not see any use of XRD and PL analyses? Provide the specific answer and identify where the authoirs have made the change in the revised manuscript? The manuscript is not acceptable in the present condition because the results are not well explained and add nothing novel to science.
5- The following comment is not addressed. "For clarity, provide a clear and concise scalebar in the SEM micrographs. Also, the relevant discussion in the manuscript draft is very unclear and vague. provide the relevant discussion in the draft with all the appropriate references and each and every aspect of the SEM micrograph should be discussed. What does the yellow, red and blue highlights mean in the Figures?"
6- The following major comment is not addressed "How does the crystal defects and career concentration affect the ZnO nanowire polarity and piezoelectric performance?".
Author Response
Kindly check the attached file
